# Delivery Systems in Ocular Retinopathies: The Promising Future of Intravitreal Hydrogels as Sustained-Release Scaffolds

**DOI:** 10.3390/pharmaceutics15051484

**Published:** 2023-05-12

**Authors:** Diana Rafael, Marcelo Guerrero, Adolfo Marican, Diego Arango, Bruno Sarmento, Roser Ferrer, Esteban F. Durán-Lara, Simon J. Clark, Simo Schwartz

**Affiliations:** 1Drug Delivery & Targeting, Vall d’Hebron Institut de Recerca (VHIR), Universitat Autònoma de Barcelona (UAB), 08035 Barcelona, Spain; diana.fernandes_de_so@vhir.org; 2Centro de Investigación Biomédica en Red de Bioingeniería, Biomateriales y Nanomedicina (CIBER-BBN), Instituto de Salud Carlos III, 28029 Madrid, Spain; 3Functional Validation & Preclinical Research (FVPR), 20 ICTS Nanbiosis, Vall d’Hebron Institut de Recerca (VHIR), Universitat Autònoma de Barcelona (UAB), 08035 Barcelona, Spain; 4Bio & Nano Materials Lab, Drug Delivery and Controlled Release, Departamento de Microbiología, Facultad de Ciencias de la Salud, Universidad de Talca, Talca 3460000, Chile; marcelo.guerrero@utalca.cl (M.G.); amarican@utalca.cl (A.M.); eduran@utalca.cl (E.F.D.-L.); 5Center for Nanomedicine, Diagnostic & Drug Development (ND3), Universidad de Talca, Talca 3460000, Chile; 6Instituto de Química de Recursos Naturales, Universidad de Talca, Talca 3460000, Chile; 7Group of Biomedical Research in Digestive Tract Tumors, Vall d’Hebron University Hospital Research Institute (VHIR), Universitat Autònoma de Barcelona, 08035 Barcelona, Spain; darango@irblleida.cat; 8Group of Molecular Oncology, Biomedical Research Institute of Lleida (IRBLleida), 25198 Lleida, Spain; 9i3S-Instituto de Investigação e Inovação, Saúde Universidade do Porto, Rua Alfredo Allen 208, 4200-135 Porto, Portugal; bruno.sarmento@i3s.up.pt; 10Clinical Biochemistry Group, Vall d’Hebron Hospital, 08035 Barcelona, Spain; roser.ferrer@vallhebron.cat; 11Department for Ophthalmology, University Eye Clinic, Eberhard Karls University of Tübingen, 72076 Tübingen, Germany; 12Institute for Ophthalmic Research, Eberhard Karls University of Tübingen, 72076 Tübingen, Germany; 13Lydia Becker Institute of Immunology and Inflammation, Faculty of Biology, Medicine and Health, University of Manchester, Manchester M13 9PL, UK

**Keywords:** hydrogels, stimuli-responsive, thermo-responsive, retinopathies, delivery systems, intravitreal delivery

## Abstract

Slow-release delivery systems are needed to ensure long-term sustained treatments for retinal diseases such as age-related macular degeneration and diabetic retinopathy, which are currently treated with anti-angiogenic agents that require frequent intraocular injections. These can cause serious co-morbidities for the patients and are far from providing the adequate drug/protein release rates and required pharmacokinetics to sustain prolonged efficacy. This review focuses on the use of hydrogels, particularly on temperature-responsive hydrogels as delivery vehicles for the intravitreal injection of retinal therapies, their advantages and disadvantages for intraocular administration, and the current advances in their use to treat retinal diseases.

## 1. Introduction to Retinal Diseases

Retinal diseases include a wide range of disorders affecting the retina, the light-sensitive tissue at the posterior of the eye. Retinal degeneration, leading to blindness, is a major disease category affecting millions of people worldwide [1]. While not lethal on its own, vision loss is known to have a major impact on the physical and mental health of patients, representing one of the top fears of patients in a recent global survey (www.theharrispoll.com (accessed on 15 April 2022)). Age-related macular degeneration (AMD) and diabetic retinopathy (DR) are among the most common diseases that cause visual impairment affecting the posterior segment of the eye. The prevalence of moderate to severe visual impairment and blindness worldwide stands at 285 million people, with 65% of visually impaired and 82% of all blind people being 50 years and older, according to the WHO. In the aging population, the prevalence of visually debilitating conditions represents 10% in AMD, and 2% in DR (https://www.who.int/publications/i/item/9789241516570 (accessed on 1 April 2023)).

AMD is a multifactorial disease with a myriad of risk factors [2]. The disease is characterized by the progressive deterioration of the macula, the central part of the retina in the back of the eye, which is responsible for detailed visual acuity [3]. The late-stage disease is divided into two main subtypes: dry AMD, characterized by islands of defined cell death referred to as geographic atrophy (GA); and wet AMD, characterized by the formation of new leaky blood vessels in a process known as choroidal neovascularization (CNV). While some success has been achieved in treating the wet form of AMD with anti-angiogenic agents, very limited options remain for a patient suffering from dry AMD, which was considered an untreatable disease until only very recently. The other major retinal disease, DR, arises as a complication of diabetes and can lead to vision loss if left untreated. DR is caused by damage to the blood vessels in the retinal tissue itself, which can leak and cause the formation of scar tissue [4]. There are four stages of PR, ranging from mild, non-proliferative DR to proliferative DR as the most advanced form of the disease, which can also result in retinal detachment. The risk of developing diabetic retinopathy depends on the duration and type of diabetes mellitus [5]. In patients having type 1 diabetes for 14 years or longer the risk is 96%, and in those having type 2 diabetes for 14 years or longer the risk is 72%. Diabetic macular edema (DME) is a complication of DR where the leakage of fluid from blood vessels in the macular region of the retina causes swelling and thickening of the retinal tissues through inflammation.

Although both AMD and DR have differing underlying mechanisms and risk factors, they do share some common similarities, as abnormal blood vessel formation, inflammation and the activation of the complement system affecting the retina are common features of both these diseases (particularly the central complement protagonists C3 and C5) [6]. While the exact underlying mechanisms remain subtly different, and in some cases not entirely elucidated, there is a renewed focus on therapeutically targeting the posterior of the eye in an attempt to prevent excessive blood vessel leakage/growth (e.g., using anti-VEGF compounds) and the inappropriate activation of the complement system (e.g., using anti-C3 or anti-C5 compounds) [7].

Furthermore, inherited retinal diseases (IRDs), such as retinitis pigmentosa (RP), are caused by genetic mutations that affect the function or structure of the retina [8]. RP is a group of rare disorders that affect the retina causing gradual vision loss. Symptoms often begin with difficulties with night vision (nyctalopia) and progressing to the loss of the mid-peripheral visual field resulting in tunnel vision. Although total blindness is uncommon, visual decline is relatively quick to include the far peripheral field, increasing the tunnel vision effect in patients. Other inherited retinal diseases include Stargardt disease and Leber congenital amaurosis. Given their genetic nature, IRDs are leading candidates for treatment by gene therapies, which aim to replace faulty genes to eliminate the molecular consequences of mutations associated with driving the various diseases.

With so many millions of people suffering from retinal diseases, it is perhaps of little surprise that they are the focus of a number of therapeutic initiatives attempting to slow progression, or even repair the associated tissue damage. Anti-VEGF drugs have led the way to treating the incidence of CNV in both wet AMD and DR (see Table 1), but often it is the method of delivery of these and future therapies that causes serious considerations. In order to better understand the hurdles faced in delivering a novel therapeutic, we need to understand the naturally occurring barriers that are designed to keep external agents out of the eye.

In this review, we will discuss the clinical and technical challenges of intraocular therapy to treat retinal diseases and the current drawbacks of the available treatments. We will also discuss the advantages and disadvantages of the existing administration routes, the pharmacokinetic challenges of intravitreal drug delivery targeting the retinal tissue, and the impact of the ocular biological barriers. Moreover, we will discuss intravitreal sustained-release delivery systems and particularly, the use of hydrogels and stimuli-responsive hydrogels as promising scaffolds to increase long-term treatment efficacy and reduce morbidity in the clinical management of retinopathies.

## 2. Ocular Barriers, a Hurdle to Overcome

Physical ocular barriers, such as the corneal epithelial barrier and the blood retinal barrier (BRB), play critical roles in protecting the eye from external and systemic insults and are vital for maintaining the functioning integrity of the eye. Additionally, the sclera, with its elastin fibers and collagenous fibrils maintaining an avascular, tightly packed extracellular matrix (ECM), provides a tough supporting outer layer enclosing the eye that varies between 0.5 mm and 1 mm in thickness [9]. The effectiveness of these barriers is directly proportional to the homeostasis of the eye and all the cell types within. Their disruption, destruction, or even natural age-related changes are known to contribute to a range of eye diseases. These include dry eye disease [10], corneal ulcers [11], DR [12], and AMD [13]. Furthermore, the integrity of these physical barriers and their highly effective role in excluding unwanted materials and external insults into the eye also pose a significant hurdle to ocular drug delivery. Even anatomical features within the eye itself can act as distinctive barriers, dictating the passage of materials. For example, the inner limiting membrane (ILM) separates the retinal tissue layers from the vitreous humor and is widely recognized as a major barrier to administered ocular therapies (Figure 1) [14]. The ILM is full of characteristic heparan sulfate sulfation sequences, particularly 3-O and 6-O-sulfation [15], which has implications for the binding and regulation of proangiogenic growth factors, such as VEGF and FGF2 [16,17]. This is believed to contribute to the prevention of aberrant blood vessel growth from the neurosensory retina into the vitreous [15]. Even a patient’s tear film, more complex than initially appreciated [18], can pose a significant barrier to the delivery of drugs, their durability, and their retention [19].

The cornea is comprised of five separate layers including the epithelial barrier, Bowman’s layer, stroma, Descemet’s membrane, and the corneal endothelium (Figure 1). The corneal epithelial barrier is the outermost layer of the cornea that directly faces the outside environment and is a major physical barrier that is under constant bacterial and environmental insult [20,21]. It is composed of the corneal epithelium, which is a layer of tightly packed, non-vascularized cells that provide a permeable barrier to the underlying tissues. The presence of tight junctions between, and active transporters within, the cells help ensure a regulated influx and efflux of molecules, including drugs, across the cornea [22]. The corneal epithelium sits on top of Bowman’s layer, a collagenous-rich ECM about 14 μm-thick that separates the epithelium from the corneal stroma. This stromal layer comprises regularly arranged collagen fibers and sparsely distributed, interconnected keratocytes [23]. This transparent layer contains almost 200 layers of mainly type I collagen fibrils varying in diameter throughout specific regions of the tissue [24]. The corneal stroma is sandwiched by another ECM, referred to as Descemet’s membrane, which is a thin, acellular ECM comprising mostly collagen type IV fibrils. Finally, adjacent to Descemet’s membrane is the 5 μm-thick corneal endothelium, which is a mitochondria-rich cell monolayer. This cell layer is in direct contact with the aqueous humor and plays a significant role in the regulation and transport of fluid and nutrients from the aqueous humor to the corneal stroma. Similarly, the blood aqueous barrier controls the exchange of substances from the plasma directly to the aqueous humor itself [25]. The barrier comprises non-pigmented ciliary epithelial cells from the ciliary body and the pigmented endothelial cells of the iris. The tight junctions between each cell in the respective monolayers ensure a strict regulation of molecules that may pass and pose a formidable barrier for drug transport [26].

Aside from environmental insults, the major anatomical feature that prevents the ingression of unwanted materials, cells or foreign factors from blood circulation into the ocular space is the BRB. The BRB comprises two major physiological barriers, which are termed the inner BRB, consisting of tight junctions between endothelial cells, and the outer BRB, consisting of tight junctions between retinal pigment epithelium (RPE) cells and the selective permeability of their underlying extracellular matrix, called Bruch’s membrane (Figure 1). This complex dual vascular system supplies oxygen and nutrients to the neural retina, which has a higher consumption of oxygen per unit weight of tissue than any other part of the human body [27]. The fenestrated blood vessels underlying Bruch’s membrane, called the choriocapillaris, supply nutrients to the cells of the outer retina, while capillaries from the central retinal artery do the same to the cells of the inner retina. These two vascular beds, although designed for the same purpose, are physiologically quite different. The capillaries of the inner BRB have a continuous endothelium with a barrier function [28] and are organized into two layers within the inner retinal tissue; the outer layers of the retina are avascular. Conversely, the endothelium of the choriocapillaris is fenestrated and permeable, although not without regulation. Despite the fact that the choriocapillaris is fenestrated, it does not allow the free transport of large macromolecules out of circulation; rather, this is regulated by cellular processes including caveolae-mediated transcytosis [29].

## 3. Current Principal Administration Routes for Ocular Therapies

Given the presence of so many effective barriers in the human eye, ocular drug administration is not entirely straightforward. Indeed, a number of different administration routes have been employed to allow therapies not only to reach the required ocular tissues, but at the required doses to increase therapeutic efficacy. Of the variety of ocular drugs currently either in development or clinically available, their application can be divided into five routes of administration: topical, subconjunctival, intravitreal (IVT), and subretinal or suprachoroidal injections, which are primarily used for the delivery of gene therapies.

Topical administration was initially predominantly used to treat conditions affecting the cornea and anterior chamber, given that it is convenient, non-invasive, and ideal for the treatment of external eye conditions. Nevertheless, eye drop administration is not as straightforward as perhaps one would assume, with the tear film and its clearance posing a major hinderance for topically administered drugs to achieve the required dosage and retention time [30]. Indeed, a large portion of any topical drug is usually washed away from the corneal surface in 2–3 min following application, significantly reducing the drug contact time with the corneal surface. However, conditions such as dry eye disease and corneal keratitis can be treated quite effectively with the topical administration of drugs directly onto the cornea [31]. Interestingly, some topically administered therapies can penetrate into the interior layers of the eye, not only through direct corneal absorption, but also through non-corneal absorption routes, such as by crossing the anterior sclera into the uveal tract, with the drug reaching the iris–ciliary body, bypassing the aqueous humor [32]. This mode of transport has been demonstrated with insulin administration in rabbits [33,34]. This phenomenon has invigorated the discussion around the topical administration of medicines to treat diseases affecting the posterior of the eye, such as AMD and DR [35,36], although whether such an approach will prove clinically efficacious remains to be determined.

Subconjunctival administration involves the injection of therapies into the space between the sclera and the conjunctiva, and are often used for administering drugs for treating posterior segment eye diseases such as uveitis [37]. However, subconjunctival injections are difficult to administer and not without their risks, where injections of steroids placed over a diseased cornea or sclera can cause thinning, temporary pain, and possibly a rupture at the site of the injection [38]. IVT injections are another method of delivering drugs into the eye for targeting diseases affecting the eye’s posterior segment. The regular delivery of anti-VEGF drugs through IVT injections has revolutionized the treatment of choroidal neovascularization (CNV) secondary to AMD, rendering this disease treatable via a relatively safe administration option [39]. As long as the drug is pharmacologically active for a reasonable period of time without limitations through breakdown or clearance, anti-VEGF injections can be administered on a monthly or every two-month basis. However, despite the success of such treatments, patient adherence remains a challenge, especially over longer periods of time (i.e., >2 years), which can lead to a decrease in therapeutic efficacy [40,41]. Indeed, the recent global COVID-19 pandemic caused increased anxiety and a marked reduction in adherence to clinical appointments for regulator anti-VEGF treatments, leading to a measurable decrease in associated visual acuity in those patients not adhering to strict administration protocols [42,43,44]. One method of drug administration designed to overcome these regular clinical visits are implantable devices, where drug-eluting implants or a sustained-release device can be inserted into the eye [45]. Implantable devices can provide sustained drug delivery over an extended period of time and can be used for the treatment of chronic eye conditions, such as glaucoma. Indeed, much interest has been generated around using such refillable devices to deliver anti-VEGF therapies in a direct response to dealing with the patient compliance issue discussed above [46]; requiring a refill once or twice a year is believed to represent a much more palatable option for elderly patients.

A relatively recent exciting field in ocular therapies is gene therapy: the delivery of a genetic payload either to replace a faulty inherited gene in a specific cell type or to promote the secretion of a locally produced drug. The first ever ocular gene therapy approved by the FDA was Luxturna (Voretigene neparvovec) for the treatment of IRD driven by biallelic *RPE65* mutations (such as in Leber’s congenita amaurosis [47]). Luxturna contains *RPE65* cDNA encapsulated in adeno-associated virus serotype 2 (AAV2) and is delivered to RPE cells in a patient’s eye by subretinal injection. The transduction of RPE cells by the AAV2 allows the incorporation of the “good” *RPE65* cDNA in an attempt to counteract the faulty genes’ contribution to the patient’s progressive blindness. Subretinal injections rely on inducing an iatrogenic retinal detachment: the controlled separation of the photoreceptor cells from their underlying RPE cell monolayer, between which the gene therapy is injected and absorbed into the surrounding cell layers [48]. Although a surgical procedure, this technique is associated with minimal trauma and early retinal structural and functional recovery, suggesting a good safety profile overall [49], but is, of course, not without potential problems [50]. One major advantage of subretinal gene therapy is that it is generally considered a “one-and-done” approach, which is to say that one application is expected to be sufficient to have continued efficacy over very long periods of time. However, it is probably pertinent to note here that with gene therapies being so early in their development, it is not entirely clear how long efficacy will last, but as of 2022 we have seen 10 years of continued Luxturna expression in animal models and 7.5 years in humans [51]. Remarkably, the long-term episomal (not integrated into the host cell genome) persistence of transcriptionally active recombinant AAV genomes has been reported, as recently reviewed by Leroy BP et al. [51]. Beyond the replacement of faulty genes, subretinal gene therapy is also being employed to deliver transgene payloads transcribing secreted soluble proteins for treating other retinal diseases such as AMD. Although still in clinical trial stages, a number of drugs targeting the complement system, which is a potent part of a host’s innate immune system and both genetically and biochemically associated with driving AMD, are being considered [13]. Furthermore, several trials are investigating the gene therapy delivery of drugs for achromatopsia, choroidema, RP, and even anti-VEGF against CNV [52].

An alternative to the subretinal gene therapy delivery is suprachoroidal gene therapy delivery. Although the principle of the gene therapy itself is the same, the region of the eye targeted and the execution of the administration are quite different. Here, a drug is delivered to the choroidal side of the outer blood/retinal barrier in the posterior of the eye using a microneedle or microcannula, where the drug is delivered to the space between the sclera and choroid. The drug is distributed circumferentially through the suprachoroidal space towards the posterior of the eye. This approach is a direct response to the difficulties in delivering therapies across the blood/retinal barrier itself. However, dosing may prove to be problematic, especially with dilution upon application and the fact that the choroid and choriocapillaris has some of the highest blood flows in the human body [53,54]. Both subretinal and suprachoroidal gene therapies are surgical procedures and suffer from a potential toxicology issue, since many patients already carry naturally occurring anti-AAV antibodies, leading to an inflammatory response to the therapy delivery system itself [55]. This can be controlled in many instances but relies on the potency of the transgene being sufficiently high, allowing for low doses of AAV to be administered while still obtaining a functioning result. One final problem with gene therapies is their affordability. Luxturna treatment was originally priced in 2018 at $425,000 per eye [56], and although the costs of developing and manufacturing gene therapies has decreased since then, this still remains a barrier to reimbursement in markets with publicly funded healthcare systems [57].

## 4. The Current Landscape of Approved Treatments for Retinal Degeneration

Despite the efforts being applied to the development of new ocular therapies for treating retinal disease, there are currently only a limited number that are actually approved and in clinical use, which utilize different administration routes (Table 1). For the most part, IVT injections are by far the most commonly accepted method for the delivery of therapies into the eye. These include humanized monoclonal antibodies such as Bevacizumab (Avastin^®^, Roche, Indianapolis, IN, USA, used off label. IgG), Brolucizumab (Beovu^®^, Novartis, Basel, Switzerland. FDA approval: October 2019. A single chain humanized antibody fragment, scFv), Faricimab (Faricimab-svoa^®^; Vabysmo^®^, Roche, USA. FDA approval: January 2022. The first bispecific IgG1 antibody that targets both the factor A of VEGF and Angiopoietin 2), and rRnibizumab (Lucentis^®^, Novartis, Switzerland. FDA approval: August 2012. A monoclonal antibody fragment, Fab). There are also aptamers such as Pegaptanib (Macugen^®^, Eyetech Pharmaceuticals/Pfizer, New York, NY, USA. FDA approval: December 2004. A pegylated modified oligonucleotide that binds and inhibits the VEGF isoform VEGF165), and fusion proteins such as Aflibercept (Eylea^®^, Bayer, Leverkusen, Germany. FDA approval: November 2011. Produced as recombinant protein in CHO cells and composed by fragments from the extracellular domains of VEGFR1 and VEGFR2 and the constant region (Fc) of human IgG1). Finally, there is conbercept (Lumitin^®^, Chengdu Kanghong Biotech, Chengdu, China. China state FDA (CFDA) approval: December 2013. Composed of the second Ig domain of VEGFR1 and the third and fourth Ig domains of VEGFR2 and the constant region (Fc) of human IgG1). However, even though conbercept showed promising results regarding its efficacy and safety in different clinical trials conducted in China in patients with neovascular AMD (phase II AURORA and LAMP trials, and phase III PHOENIX trial), it unfortunately failed to reach the primary endpoints (mean change from baseline in best corrected visual acuity (BCVA) at Week 36) of the phase III clinical trials PANDA-1 (NCT03577899) and PANDA-2 (NCT03630952) in AMD patients. Because of this, both trials prematurely ended in June 2021 and FDA approval was not granted (and hence we do not include conbercept in Table 1).

More recently, the complement inhibitor Syfovre™ (Pegcetacoplan, Apellis Pharmaceuticals, Waltham, MA, USA), a PEGylated peptide that targets the complement C3 protein, has been approved by the FDA (February 2023) for the treatment of geographic atrophy secondary to AMD. The efficacy and safety of the IVT administration of Pegcetacoplan was successfully evaluated in the phase III clinical trials DERBY (NCT03525600) and OAKS (NCT03525613) in a dose regimen IVT injection administration of every 25 to 60 days.

Another method to treat unwanted vascular growth in the retina that does not require regular injections is laser photocoagulation surgery, where concentrated lasers are used to destroy and seal abnormal blood vessels [58]. The treatment itself is effective but can be painful, and by definition the technique intentionally causes scaring of the retinal tissue itself and can lead to the creation of new blind spots in the patient’s vision [59], and is less and less commonly used in clinics.

Finally, corticosteroids are another class of drugs that have been approved for the treatment of inflammation related to retinal diseases, such as uveitis and diabetic macular edema. These anti-inflammatory drugs are very effective at reducing the associated inflammation and retinal swelling and can be administered as either injections or longer-term slow-release implants [60,61]. Corticosteroid implants such as Ozurdex (Dexamethasone) or Iluvien (Fluocinolone acetonide) have been approved by regulatory agencies in the US and Europe. Despite good efficacy at reducing swelling of the retinal tissues, neither of these drugs addresses the fundamental underlying reasons for the inflammation in the first place, so they are often used in conjunction with other therapies.

## 5. Facing the Drawbacks of Intravitreal Treatments

Despite numerous efforts during the past decade, the delivery of proteins and drug-payloads to the retina still remains a significant challenge. Among the different administration routes, IVT injections are the most frequently used in clinics to target the posterior segment of the eye since injected drugs and proteins can diffuse within the vitreous in the vicinity of the retina [62,63,64,65] (see Table 1). Unfortunately, current treatments still require too frequent IVT injections. This not only hampers the adherence of patients to the treatment but also substantially increases the risks of adverse reactions and undesirable secondary effects including retinal detachment, intraocular infections, cataract formation, hemorrhages and increased intraocular pressure, among the most important [66]. For example, anti-VEGF biologics have IVT half-lives in the range of one week and must be injected every 4–6 weeks, resulting in about 100-fold fluctuations of ocular drug concentrations during chronic treatments [67].

Moreover, the adequate IVT diffusion and clearance of therapeutic payloads are fundamental pharmacokinetic parameters that strongly influence the efficacy of a treatment and importantly, their dose administration regimen. Many preclinical studies have been performed in animal and human eyes to optimize the IVT pharmacokinetic profiles of these inhibitors in order to maximize their therapeutic windows and their treatment efficacy. These studies show that the IVT half-life of current VEGF inhibitors is in the range of 2 to 7 days in animal eyeballs from rabbits (New Zealand white [64,68,69,70,71,72,73,74,75], Chinchilla rabbits [76] and Dutch-belted rabbits [62,77,78,79]) and macaques (Cynomolgus monkey [73,80,81,82], Rhesus monkey [83] and Owl monkey [84]), and in the range of 5 to 13 days in human eyes [85,86,87]. Moreover, a recent FDA clinical pharmacology review of Brolucizumab (26 kDa MW) reported IVT half-lives of 2.08 and 2.94 days, respectively, in Cynomolgus monkey and New Zealand rabbits, and 5 ± 2 days in human eyes, as reported in the RTH258-E003 clinical trial [73].

Nowadays, most therapeutic payloads fail to show adequate bioavailability, pharmacokinetics (PK), and therapeutic efficacy because diffusion across the biological barriers of the eye and through the vitreal fluid is unsuccessful. These approaches also have the drawbacks of low efficiency, off-target delivery, side effects, and a short lifespan [66,88,89,90,91,92]. Moreover, a limited number of formulations and administration routes (e.g., eye drops, subconjunctival, subretinal, subchoroidal and IVT injections) are available for the delivery of therapeutic components to the eye (e.g., drugs, protein, peptides, genes, nanoparticles). However, challenges remain on how to improve the efficacy of drug delivery and minimize side effects to target the posterior segment of the eye, where several physiologic and biologic barriers exist that must be overcome, and where a long IVT half-life of therapeutic compounds is required.

## 6. Understanding the Pharmacokinetic Challenges of Intravitreal Drug Delivery

The vitreous humor is an open, three-dimensional hydrophilic network formed by hyaluronic acid (HA) (0.5%), collagen (0.5%) and water (99%), with a mesh size of about 500 nm [66,93,94]. The diffusion of small, soluble proteins under 10 nm in size and small molecules, in particular lipophilic molecules, is usually not restricted by the vitreous mesh, which increases their bioavailability in the retina. Conversely, they often show fast clearance rates from the posterior segment of the eye ranging from 1 to 10 h, reducing their long-term effects and forcing frequent IVT administrations as a major drawback. In this regard, there are two clearance mechanisms in the vitreal cavity: on the one hand, the anterior route followed by most drug payloads, where drugs diffuse through the vitreous and leave the cavity using the aqueous humor outflow, and on the other hand, the posterior route that implies crossing the BRB and reaching systemic circulation [95], used only by a minor fraction of drug payloads. On the contrary, polymer-based and nanoparticulated delivery systems, as well as positively charged proteins and molecules, can suffer restrictive IVT diffusion because of their size and/or charge interactions with the negatively charged HA mesh [96,97], considerably reducing their retinal bioavailability and thus their therapeutic efficacy. Because of these, the vitreal clearance of most proteins and macromolecules is in the range of several days [98,99]. Furthermore, most therapeutic drugs (e.g., anti-VEGF proteins) are large molecules with low permeability across the BRB, and the drug payload, release and material properties are often suboptimal. Furthermore, the use of delivery platforms often presents difficulties regarding their controlled release and degradation time, the formation of toxic degradation sub-products, and sub-optimal rheological features of the vitreal fluid. While anionic and neutral nanostructures (e.g., 100 nm) diffuse well in the vitreous, cationic particles bind to the polyanionic hyaluronic acid of the vitreous mesh and diffuse 100–1000 times slower [89]. Moreover, delivery systems often provide a burst-type effect that is not capable of sustained efficacy in the mid-to-long term, requiring repeated injections in short time periods [100,101].

Additionally, most drug payloads are not capable of delivering long-term efficacy primarily because of problems in overcoming the biological barriers in the eye, lacking appropriate bioavailability in the target sites, or suffering rapid clearance. In fact, for retinal delivery the carrier needs to also permeate across the ILM at the vitreous–retina interface (Figure 1). The ILM has a smaller mesh size (50–100 nm) than the vitreous (≈550 nm), where possibly only the smallest particles (<100 nm) will enter the retina. Of note, the vitreal clearance rate in human patients is 1.4 times higher than that in rabbits and the distribution volume is 3-fold higher; therefore, half-life times in the human vitreous double as compared with that of the rabbit [66,89,98,102]. A very important study by Kim et al. was recently performed in rabbit eyeballs to define prediction models of the intraocular pharmacokinetics of micro- and macromolecules based on their physicochemical properties [90]. The best-fit models show a strong correlation between the molecular weight (MW) and the IVT half-life in macromolecules (defined as MW > 900 Da), whereas both the MW and lipophilicity are determinant factors in micromolecules (defined as MW ≤ 900 Da). This study concluded that the IVT half-life can be predicted based on the MW and lipophilicity as major molecular physicochemical properties and suggested that increasing the MW while reducing lipophilicity can prolong the IVT half-life of small drugs, while increasing the MW can be the single most important determinant when considering large biologicals. Nonetheless, increasing the MW might be challenging for the delivery of biologics to the retina since a larger MW also causes slower diffusion in the vitreous cavity and reduced clearance through both the anterior and posterior pathways, as well as hinders the capacity to cross biological barriers and to permeate membranes (i.e., ILM, the retinal pigment epithelium, and the BRB) [67,91,92]. In this context, KSI-301 (Tarcocimab tedromer^®^, KODIAK Sciences, Palo Alto, CA, USA) is an anti-VEGF compound of 950 kDa composed by a monoclonal antibody conjugated to a large biopolymer backbone. A phase I study (NCT03790852) performed in patients with neovascular AMD, DME and retinal vein occlusion showed safety and visual improvement with 6-month IVT injection intervals [103]. phase II/III human clinical trials in patients with neovascular AMD treated every 3 months showed promising results, according to a KODIAK announcement in Feb 2022 (DAZZLE, NCT04049266) [104]. This is despite KSI-301 failing to meet the primary endpoint of achieving non-inferiority to Aflibercept and some concerning signs of intraocular inflammation being detected. New data are expected in 2023 from the ongoing trials BEACON (NCT04592419), GLEAM (NCT04611152), GLIMMER (NCT04603937), GLOW (NCT05066230) and DAYLIGHT (NCT04964089) with different dose injection regimens (www.clinicaltrials.gov (accessed on 5 April 2023)).

Recently, the use of a novel class of highly stable, engineered small proteins named DARPin as VEGF inhibitors are being investigated. DARPin proteins contain ankyrin repeat domains designed to bind and inhibit specific protein targets with high affinity and specificity [105]. Abicipar pegol is a DARPin drug with a MW of 34 kDa that is designed for the treatment of wet AMD, targeting VEGF. A phase I/II, open-label, multicenter, dose escalation trial was performed to evaluate its safety and bioactivity in patients with DME [106,107]. Unfortunately, even though promising levels above the half-maximal inhibitory concentration of bicipar pegol were detected in the aqueous humor for 8 to 12 weeks in this trial, the FDA has recently declined its use because of an unfavorable benefit–risk ratio, since a high rate of intraocular inflammation was reported in patients enrolled in the phase III CEDAR (NCT02462928) and SEQUOIA (NCT02462486) clinical trials.

## 7. Intravitreal Delivery Systems for Sustained Drug Release

### 7.1. Intraocular Implants

An alternative to prolonged IVT injections is the use of long-term intraocular implants to improve the sustained release of small drugs to the retina [45,108,109,110,111,112,113]. In fact, intraocular implants are currently the only delivery systems approved by the FDA for intraocular use. Drugs are conjugated to a polymer-based platform to allow their sustained release inside the vitreal cavity. According to the nature of the polymers used, implants can be divided into: (i) non-biodegradable implants, most of which use poly(ethylene–co-vinyl acetate) (pEVA), poly(dimethyl siloxane) (PDMS), or poly(vinyl alcohol) (PVA); and (ii) biodegradable implants mostly made of poly(lactic–co-glycolic acid) (PLGA), poly(glycolic acid) (PGA), or poly(lactic acid) (PLA) [108,109,110,111]. Whereas most biodegradable implants might sustain drug release and efficacy up to 6 months due to progressive IVT scaffold degradation and an exponential decrease in IVT drug concentration over time, non-biodegradable implants might extend drug release up to 2–3 years following a linear decrease in IVT drug concentration, while the scaffolds remain intact within the vitreous cavity.

An example of a biodegradable implant is Ozurdex^®^ (dexamethasone corticosteroid conjugated to a polymer backbone of poly(lactic–co-glycolic acid) indicated for the treatment of macular edema following branch retinal vein occlusion (BRVO) or central retinal vein occlusion (CRVO) and moreover, for the treatment of inflammation (non-infectious uveitis) of the posterior segment of the eye. The implant allows the controlled release of Dexamethasone up to 4–6 months and degrades entirely in vivo [45,112]. Other examples are Brimo DDS^®^ (a brimonidine drug delivery system in a poly(D,L-lactide) biodegradable polymer matrix), a selective α 2-adrenoceptor agonist currently approved in the United States and Europe for the treatment of open-angle glaucoma and ocular hypertension, which is currently in phase III clinical trials for the treatment of patients with geographic atrophy secondary to neovascular AMD [113,114]. Both from Allergan Inc. (Irvine, CA, USA). Moreover, the FDA has recently approved Durysta^®^ (Bimatoprost, a prostaglandin analogue in a biodegradable polymer matrix consisting of a poly(D,L-lactide), poly(D,L-lactide–co-glycolide), poly(D,L-lactide) acid end, and poly(ethylene glycol) 3350. Allergan; FDA approval March 2020) to reduce the intraocular pressure caused by open-angle glaucoma, thereby preventing further retinal damage.

Furthermore, Retisert^®^ (Fluocinolone acetonide corticosteroid. Bausch & Lomb, Rochester, NY, USA), an implant consisting of a tablet within a silicone elastomer cup with a release orifice and a polyvinyl alcohol membrane (RETISERT [Package Insert]. Rochester, NY, USA: Bausch & Lomb Incorporated.; May 2019. 2. U.S. Food & Drug Administration. https://www.accessdata.fda.gov/drugsatfda_docs/nda/2005/021737s000TOC.cfm, accessed on 4 October 2017) and Illuvien^®^ (Fluocinolone acetonide corticosteroid in a solid matrix made of silicone, polyimide and poly(vinyl alcohol). Alimera Sciences Inc., Aldershot, UK) are non-biodegradable implants approved by the FDA to treat diabetic macular edema, macular edema secondary to retinal vascular occlusion, and posterior uveitis [115,116,117]. Another non-biodegradable implant is Vitrasert^®^ (Ganciclovir in a silicone-based matrix from Bausch & Lomb, Rochester, NY, USA) for the treatment of retinitis caused by cytomegalovirus.

Unfortunately, the use of implants involves invasive procedures and requires surgical implantation. In addition, non-biodegradable implants also require surgical removal after drug release is completed. For this reason, they carry a higher risk of infection, an associated rise of intraocular pressure, and an increased incidence of post-operative cataracts [108]. Furthermore, most intraocular implants are not designed to provide a sustained release of intraocular therapeutic proteins because of their large size (high MW), hydrophilicity, and subsequent protein stability, which hampers their suitability for the administration of the current VEGF inhibitors [118].

### 7.2. Hydrogels for Intravitreal Drug Delivery

New delivery systems are required to ensure the long-term sustained release of therapeutic payloads to prolong injection intervals and to reduce morbidity risks and side effects while targeting the retina. Among them, the use of injectable aqueous biodegradable gels formulated using hydrophilic polymers (hydrogels, HGs) and of in situ-forming HGs (stimuli-responsive HGs, SRHGs) are gaining particular attention as delivery vehicles for intraocular treatments as long-term release platforms for drugs and proteins [89,119,120,121,122,123,124] (Figure 2). Generally speaking, HGs are water-swollen tridimensional networks made of polymers that are produced by the conjugation/reaction of one or more monomers. Moreover, HGs contain high amounts of water due to the presence of numerous hydrophilic functional groups, allowing the entrapment and/or conjugation of high payloads of proteins and drugs. In addition, HGs can be adapted through the modification of certain properties such as their biodegradability, swelling degree, pore size, permeability, viscoelasticity and hydrophilicity. The selection of monomers and polymers, their concentration, and the degree of cross-linkage and the chosen linkers directly impact the physicochemical attributes of the HGs and have a strong influence on their degradation profile and the release of protein payloads. As an example, hydrophilicity and cross-linking density can alter the pore size and protein diffusion rates, which strongly influence drug and protein release by changing the swelling degree of the HG. Moreover, cross-linking density can also be tailored by modifying the molecular weight, concentration and architecture of the chosen polymers, which together with the type of selected biodegradable linkers directly affects the degradation rate of the HG [89,119,120,121].

According to the nature of the polymers used to produce the HGs, polymers can be classified into three main categories: synthetic polymers such as poly(N-isopropylacrylamide) (PNIPAAm), poly(lactic–co-glycolic acid) (PLGA), poly(ethylene glycol) (PEG) and its derivatives poly(ethylene glycol) diacrylate (PEGDA) and poly(ethylene glycol) methacrylate (PEGMA), and poly(caprolactone) (PCL); non-synthetic (natural) polymers such as hyaluronic acid (HA), alginate, chitosan, cellulose, dextran and silk; and hybrid polymers, which contain a mix of synthetic and non-synthetic polymers [89,122,123]. The presence of hydrolysable linkages within the polymers makes these HGs biodegradable by chemical and/or enzymatic hydrolysis. Furthermore, HGs can also be classified according to the type of cross-linking as covalent and non-covalent cross-linked HGs [124]. While non-covalent forces (hydrogen bonds, electrostatic charges and hydrophobic interactions) are weak and reversible depending on environmental conditions, covalent links are based on strong chemical binding between polymer chains (small cross-linking molecules, polymer-to-polymer chemical reactions). Of note, non-covalent cross-linked HGs are more difficult to tune up as their interaction and reactivity with the environment as well as their biodegradability and payload release are far from predictable [124,125,126,127,128,129,130]. On the other hand, covalent cross-linked HGs require chemical reactions that might cause the inactivation of bioactive molecules and protein payloads. In addition, the use of cross-linking molecules can also increase their in vivo toxicity and strongly affect protein release.

### 7.3. Release Kinetics of Hydrogels at the Vitreal Cavity

In these regards, the release of bioactive molecules such as anti-VEGF proteins into the vitreal cavity can be achieved either by the direct diffusion of the protein from the HG (diffusion-controlled) and/or by the progressive degradation of the gellified structure (degradation-controlled) [125,126,127]. Both mechanisms are often overlapped. In fact, the release kinetics of most HGs show an initial burst-type effect (burst-release phase) produced just before the onsite cross-linkage of the gelation state is completed. This phase can be responsible for releasing up to 50% of the total protein payload into the vitreous [121,128,129]. Thus, a large percentage of the payload can experience a faster clearance than initially desired. A second phase characterized by a slow, diffusion-controlled release of the remaining protein payload follows. This diffusion depends on the pore size of the HG and on the remaining payload concentration, and it is progressively affected by the chemical and/or enzymatic degradation (degradation-phase) of the polymer network. Because of the high content of water and the sponge-like architecture of the HG, this degradation takes place from both the inside and the outside of its three-dimensional structure. In time, the release of the payload decreases, as it does its concentration within the hydrophilic spaces of the polymeric network, until a final burst-release phase occurs when the collapse of the HG matrix takes place [127,130,131,132,133]. Improved fine-tuning of the attributes and rheology of the HGs is required in order to better conceal the dynamics of their progressive structural changes and degradation with the clinical demand of slow, sustained-release systems for bioactive proteins in the vitreal cavity. Nonetheless, the delivery of anti-VEGF proteins is particularly challenging because chemical treatments, heat and/or pH modifications, among other environmental factors, might strongly alter their tertiary and quaternary structure, which are determinant of their bioactivity [134]. In this regard, the porous tridimensional matrix structure of the HGs not only allows the loading of high amounts of anti-VEGF proteins but also protects them from environmental aggressions that can cause their denaturation, thus preserving their activity. Subsequently, HGs are nowadays considered ideal delivery systems for the sustained release of bioactive hydrophilic agents into the vitreal cavity, including current anti-VEGF protein treatments. Yu et al. developed a vinyl sulfone functionalized hyaluronic acid (HA–VS)-thiolated dextran (Dex-SH) in situ-forming HG for the IVT controlled release of Bevacizumab that showed good biocompatibility in rabbit eyes by binocular indirect ophthalmoscopy, full-field electroretinogram, and histology. Bioactivity of Bevacizumab was seen for up to 6 months with IVT concentrations up to 107 times higher than for the bolus injection [123]. Zhang et al. described an in situ-forming HG that undergoes gelation upon exposure to water through hydrophobic forces and physical cross-links. The HG was prepared via simple free-radical polymerization using poly(ethylene-glycol) methyl ether methylacrylate (PEGMA) and a vitamin E (Ve) methacrylate copolymer (PEGMA–co-Ve). The HGs showed a slow degradation process determined by the water content over a two-month period in vitro, with the lower-water-content gels showing the slowest degradation times [135]. Lovett et al. reported the good biocompatibility and sustained release of high doses of Bevacizumab (Avastin™) (50 μg/mL) for at least 3 months in Dutch-belted rabbits using silk-based HGs. Remarkably, the dose release concentration of Bevacizumab at day 90 was equivalent to the dose concentration of a standard single injection of 1.25 mg of Bevacizumab at day 30. Signs of HG biodegradation were detected after 3 months [122].

Another interesting example of a biodegradable HG for IVT use is OTX-IVT, an implantable biodegradable HG for the IVT sustained release of Aflibercept or Bevacizumab for up to 4–6 months, and OTX-TKI for the sustained release of axitinib, a small-molecule tyrosine kinase inhibitor with anti-angiogenic properties, for up to 12 months, for the treatment of wet AMD, designed by Ocular Therapeutix, Bedford, MA, USA in collaboration with Regeneron Pharmaceuticals, Tarrytown, NY, USA [136]. Multicentric clinical studies are underway to evaluate the safety, tolerability, and efficacy of OTX-TKI for IVT use in comparison to an on-label 8-week Aflibercept bolus injection in subjects with neovascular age-related macular degeneration (NCT03630315; NCT04989699). In September 2022, the company made public its 7-month interim report showing the good safety profile of OTX-TKI and sustained visual acuity in treated patients. Its 10-month provisional report presented in February 2023 also showed good tolerance, safety and sustained efficacy as compared with the Aflibercept bolus. Another clinical study is underway to evaluate the safety, tolerability, and efficacy of OTX-TKI in subjects with moderately severe to severe non-proliferative diabetic retinopathy (NCT05695417).

### 7.4. Stimuli-Responsive Hydrogels in Retinopathies

SRHGs are in situ gel-forming HGs considered smart materials because they modify their tridimensional structural conformation in response to different external stimuli such as pH, temperature, ionic strength, light, biomolecules, or when subjected to a magnetic field [137,138,139,140,141,142,143,144]. Because of this, it is possible to design HGs with specific rheological attributes that allow their use as injectable solutions through small gauge needles for IVT administration. This is a clear advantage over HGs that lack syringeability and have to be suspended in water prior to their administration. Accurate dosing, simple formulation processes, and easy sterilization are additional advantages to consider [145]. Furthermore, the movement of macroscopic HGs when gellified is significantly restricted or even absent in the vitreal cavity as they are larger than the average mesh size in the vitreous network of hyaluronic acid (500 nm) [93,94].

### 7.5. Temperature-Responsive Hydrogels for Intraocular Delivery

An example of SRHGs are thermo-responsive HGs (TRHGs) (Figure 2). They are designed to undergo the sol–gel phase transition in response to the existing environmental temperature of the vitreal cavity. The transition to a gel state (gelation by polymerization, self-assembly and/or cross-linking) occurs when polymers in solution are above or below the so-called lower critical solution temperature (LCST) and the upper critical solution temperature (UCST), when the solution separates into a gel phase and a solvent phase (often water) [145,146,147,148]. The use of TRHGs for intravitreal applications is under extensive study because of their good syringeability and fast in situ gelation in the vitreous. In addition, TRHGs can also be useful as tamponade agents in the vitreal cavity for the treatment/prevention of retinal detachment, as showed by Liu et al. using a urethane-based TRHG in vitrectomized non-human primates for up to 12 months [149]. Nonetheless, challenges remain regarding how to develop in situ gelling HGs that are able to sustain protein delivery for a period of months and not just days or weeks, and also regarding how to avoid/reduce the fast burst release of the protein payload to avoid toxicity and sustain efficacy.

In this context, Drapala et al. in 2011 used free-radical polymerization to develop TRHGs made of PNIPAAm cross-linked with either poly(ethylene glycol)–co-(L-lactic acid) diacrylate (PEG–PLLADA) or poly(ethylene glycol) diacrylate (PEGDA) and showed that the cross-linking density has a direct influence on protein release and degradation. Their TRHGs provided the sustained IVT release of Bevacizumab or Ranibizumab for a month [150]. These TRHGs did not show long-term adverse effects in the retina [138]. These same authors also showed in 2014 that the controlled degradation of the system was also influenced by the addition of biodegradable copolymers and other additives such as glutathione, as a chain transfer agent, for the delivery of immunoglobulin G (IgG) and the recombinant proteins Bevacizumab (Avastin^®^) and Ranibizumab (Lucentis^®^). These authors showed that increased concentrations of glutathione accelerated the degradation rate of the TRHGs and subsequent protein release, compared with TRHGs prepared without glutathione that showed complete protein release after 3 weeks. Moreover, the PEGylation of IgG also significantly reduced the protein burst release [151].

Other examples of biodegradable TRHGs for the extended release of Bevacizumab have been previously reported. Using a copolymer poly(2-ethyl-2-oxazoline)–b-poly(ε-caprolactone)–b-poly(2-ethyl-2-oxazoline) (PEOz–PCL–PEOz) HG, Wang et al. demonstrated the extended release of bioactive Bevacizumab in vitro for up to a month using a human retinal pigment epithelial cell line. The HG also had a good temperature-sensitive sol–gel phase transition and in vivo biocompatibility in the rabbit neuroretina, and showed well-preserved histomorphology and electrophysiology after 2 months of IVT injection [152]. Park et al. created a TRHG composed of poly(ethylene glycol)–poly(serinol hexamethylene urethane) (ESHU) capable of providing the sustained in vitro release of Bevacizumab for up to 4 months. Later on, Rauck et al. demonstrated the in vivo sustained release of Bevacizumab from ESHU after its IVT administration, for over 9 weeks. No toxicity was reported using a bovine corneal endothelial cell model, nor was any significant inflammatory response elicited over a 9-week period in vivo using male New Zealand white rabbit models. Remarkably, ESHU was able to maintain a 4.7 times higher intraocular drug concentration of Bevacizumab than the repeated administration of an IVT-injected bolus [153,154]. Furthermore, Xie et al. reported sustained IVT release of Avastin(R) using injectable TRHGs composed of poly(lactic acid–co-glycolic acid)–poly(ethylene glycol)–poly(lactic acid–co-glycolic acid) (PLGA–PEG–PLGA). Sustained Avastin^®^ release was seen in vitro over a period of up to 14 days. Good biocompatibility, preserved retinal function, and the extended release of Avastin^®^ were also detected in vivo for up to 2 months [155]. Similarly, López-Cano et al. recently reported the successful use of PLGA–PEG–PLGA TRHGs for the sustained release of the neuroprotective agents Dexamethasone and ketorolac to the retina, using in vitro models [156]. In 2019, Xue et al. reported promising results using multiblock poly(ether ester urethane) thermo-responsive HGs composed of poly(ethylene glycol), poly(propylene glycol) and poly(ε-caprolactone), with HMDI (1,6-hexamethylene diisocyanate) as a coupling agent. The TRHG showed good encapsulation of anti-VEGFs into polyurethane thermogel depots and moreover, that the anti-VEGF release rates were dependent on the hydrophilic–hydrophobic balance within the copolymer. Anti-VEGF release was seen in vitro for up to 40 days, and anti-angiogenic bioactivity was detected in rat ex vivo choroidal explants and in a VEGF-driven neovascularization rabbit model [157].

### 7.6. Thermo-Responsive Hydrogels Containing Nanoformulations

Whereas it has been shown that microparticles can be cleared from the vitreous in 50 days [158], the use of mixtures of TRHGs with nano- and microparticles can reduce the initial release burst of the protein payloads and allow their localized and extended release after IVT administration [159,160]. Nanogels are HGs composed of a hydrophilic polymer network and a therapeutic payload at the nanoscale. Therapeutic payloads can be conjugated and/or encapsulated into a variety of different nanoformulations (e.g., nanoparticles and polymer-based nanoparticulate systems, liposomes, solid lipid nanoparticles, dendrimers, etc.) within the nanogel, and they can be designed as SRHGs. Therefore, the influence of the addition of nano/microparticulate systems during protein release from TRHGs has been extensively studied [159,161,162].

Li et al. demonstrated that nano- and microspheres fabricated from poly(DL-lactide–co-glycolide) (PLGA) and poly(ethylene glycol)–b-poly(D,L-lactic acid) (PEGLA), were capable of sustaining the release of Bevacizumab (Avastin^®^) for 90 days in 10 mM phosphate-buffered saline (PBS). Interestingly, changes in the drug/polymer ratio also altered the protein release rates [163]. Therefore, the potential use of microspheres in TRHGs has also been explored in order to achieve the controlled and extended release of the protein payloads. In this regard, Kang-Mieler’s group showed that the addition of biodegradable PLGA microspheres loaded with the VEGF inhibitors Ranibizumab (Lucentis™) and Aflibercept (Eylea™) in a thermo-responsive PNIPAAm–PEG diacrylate (PEGDA) HG were able to achieve a controlled and extended release of the inhibitors for 6 months in vitro. The phase transition temperature of the TRHG could also be modified by adjusting the concentration of PEGDA in the cross-linking reaction [164,165,166,167]. The group also showed the in vivo efficacy and anti-angiogenic bioactivity of the released anti-VEGF payloads in a laser-induced rat model of choroidal neovascularization for up to 12 weeks [168].

Another example of an amphiphilic thermo-sensitive PLGA nanogel as an IVT delivery system was reported by Hu et al. A TRHG composed of a methoxy poly(ethylene glycol)–block poly(lactic–co-glycolic acid)–BOX (2,2′-bis(2-Oxazoline) (mPEG–PLGA–BOX) diblock copolymer was synthesized to deliver Bevacizumab for wet AMD using Rex rabbit models. In vitro sustained anti-VEGF activity was detected using human umbilical vein endothelial cells and rhesus choroid–retina endothelial cells [169,170]. Further experiments showed in vivo anti-angiogenic activity for up to 30 days in a Rex rabbit neoangiogenic model by using laser retinal photocoagulation. In addition, good cytocompatibility and preserved retinal function were observed [171].

Other similar examples have been published mixing liposomes or nanocomposites with TRHGs to extend protein release. Pachis et al. used liposomes loaded with Flurbiprofen (Flu), a non-steroidal anti-inflammatory drug (NSAID), mixed with Pluronic^®^ F127 (a polymer sensitive to temperature that gellifies at body temperature) in IVT injections. Good biocompatibility extended the intraocular release of Flu, and tissue safety was demonstrated [172]. Furthermore, Sapino et al., also using Pluronic^®^ F127, created TRHGs containing either solid lipid nanoparticles (SLNs) obtained by cold microemulsion dilution, or nanoemulsions (NEs) made with IPM as the lipid phase and Epikuron^®^200 as the surfactant (µE1-based thermosensitive nanocomposite). Both designs showed good syringeability properties and reduced initial burst release of cefuroxime, which is a cephalosporin antibiotic for the treatment of endophthalmitis, a severe IVT inflammatory disease, using a two-compartment in vitro eye flow model (PK-Eye) to estimate ocular drug clearance through the anterior aqueous outflow [173].

## 8. In Silico Modeling to Optimize HG Design

In recent decades, new tools termed “computational methods” have allowed a change in the paradigm of the established “trial and error” by the “rational design” of HGs [174]. With regard to trial and error, previously only once the HG had been prepared in the laboratory was it possible to evidence the real release efficacy of a specific therapeutic agent [175]. Indeed, a large proportion of the HGs prepared in the traditional way do not comply with the required release profiles, resulting in wasted resources and time. However, with regard to new computational methodologies, it is possible to fully optimize the design before the synthesis of the HG in order to vastly improve therapeutic release from the HG [176]. Thanks to representative computational methods such as molecular dynamic simulation [177], docking [178], semi-empirical quantum mechanical methods [179] and free-energy perturbation [180], among others, it is possible to achieve the rational design of HGs. Basically, through these tools it is possible to better understand how the structure of HGs can interact with a specific therapeutic agent such as a drug, peptide, protein, antibody, etc., at an atomic and molecular level [180,181]. With these methods, it can be possible to predict the intermolecular interactions that govern the affinity of the HG toward a specific molecule. With this information, we can adjust the drug release rate and avoid/reduce a fast burst release by modifying the HG structure at the molecular level with different building blocks that provide the required physicochemical environment for the interaction with the therapeutic agent. However, as previously mentioned, this only complies when the load of the therapeutic agent into the HG is predominated by weak interactions, for example, hydrophilic and hydrophobic interactions, van der Waals and electrostatic interactions, hydrogen bonds, and salt bridges [176,182]. The computational methods predict the intermolecular interactions between moieties (functional groups) from HG and moieties of the studied therapeutic agent [181]. Below we outline some basic computational methods.

Molecular dynamics simulation (MDS) is classified as a method of molecular mechanics. The intermolecular properties of each system (HG–therapeutic agent) can be studied through MDS methods evaluating the molecular geometry over a time period [183,184]. The concept of selective affinity considers the non-covalent interactions that maintain the HG-therapeutic agent joined for a prolonged time until its sustained release by diffusion through the matrix or HG degradation [176]. Moreover, this method allows for the random addition of a certain number of molecules (e.g., a specific number of molecules (drug) around a designed HG) and for the evaluation of the behavior of the interaction of the HG–drug complexes [185]. Furthermore, this allows setting up both the temperature and time range of the interaction in the order of nanoseconds. On the other hand, this method allows the computing of energies and forces between biomolecules in the presence of parameters such as temperature and pressure [186].

Molecular docking is classified as a method of molecular mechanics, but unlike MDS, this method is static. This technique allows the user to perform a virtual screening to dock each building block or portion that makes up the structure of the HG with a therapeutic agent, for example, a monomer, co-monomer and cross-linker. For this analysis there are different protocols and docking programs with varied characteristics [184].

The semi-empirical quantum mechanical method or the in silico calculation of interaction energies allows the user to calculate the interaction energy of molecule1–molecule2 complexes [187]. In other words, this computational method allows obtaining the best virtual HG candidates according to their affinity (expressed as interaction energy) with the specific therapeutic agent [184]. Molecule1 corresponds to a building block of the HG and molecule2 to the therapeutic agent [186], and therefore each building block in the HG/therapeutic complex must be considered one by one. As such, this could be considered as a disadvantage of this type of analysis as the complete analysis of the overall HG structure is complicated due to its huge molecular dimension.

There are few reports reporting the use of bioinformatic tools for the virtual design of HGs. Carreño et al. in 2021 carried out the rational design of injectable HGs for the sustained and localized release of doxorubicin as approach for on demand cancer therapy (see Figure 3) [185]. Pereira et al. in 2023 performed a rational design of injectable HGs as a potential depot-based drug delivery system of antimicrobial peptides to be applied in diabetic foot infections associated with multidrug-resistant bacteria [184]. Taking into account the exact chemical structure of the selected antimicrobial peptide, the HGs were designed regarding the structure of the peptide with the aim of being released from the HG according to the required management of infection. Until now, there are no studies that involve computational methods in the rational design of injectable HGs as a depot-based drug delivery system in ocular retinopathy therapies, unless the required management of pathology is known, such as the type of drug that needs to be applied and its pharmacodynamic and pharmacokinetic profile, its release profile for treatment, and routes of drug administration. In addition to the above, if we know the physiological conditions such as pH, temperature, and ionic strength and anatomical components of the intraocular environment, then it could be possible to design injectable HGs with specific chemical structures and functional groups that confer the suitable environment to generate intermolecular interactions between the HG matrix portions and therapeutic agent. Consequently, these techniques could be extrapolated to the rational development of injectable HGs applied to retinopathies.

In this review, some therapeutic agents applied in ocular retinopathies are depicted that include peptides, monoclonal antibodies, corticoids, antibiotics and endothelial growth factors, among others. All of them have known chemical structures; therefore, it could be possible to predict the best structure of an HG with appropriate physical–chemical and biological characteristics and then promote a suitable release of the specific therapeutic agent, increasing its efficacy. Thus, using these computational methods, the “on-demand” formulations based on HGs according to the specific requirements of each pathology and patient could become a reality, achieving therapies with more effectiveness and friendliness.

## 9. Current Challenges and Future Perspectives: Facing the Clinical Development of Hydrogels

Even though in situ-forming gels are highly promising as IVT delivery systems, their clinical development and industrial production still face several challenges, particularly regarding their syringeability, the control of their degradation time, constant drug/protein release and dose range maintenance, vitreal clearance and long-term efficacy, toxicity, crossing of biological barriers, and aspects related to their scalability, storage, GMP production and sterility. In fact, generally speaking, an ideal biomaterial for intraocular delivery has to comply with several requirements [89,124]. Firstly, it has to be biocompatible, safe and transparent to avoid vision interferences. Secondly, it has to be able to upload high amounts of protein payload to ensure efficacy while at the same time protecting the payload from denaturation and/or degradation (e.g., 0.5–2 mg of anti-VEGF in a maximum volume of 0.1 mL). Thirdly, it has to be easily injectable (minimally invasive) and avoid increasing the intraocular pressure. Fourthly, protein release in an efficacious dose has to be sustained up to at least two to three months to significantly reduce the burden of repeated administrations. Fifthly, it has to be progressively biodegradable to better control the release of the payload and required dosing in order to ensure prolonged efficacy, and finally, to allow easy manufacturing under GMP ensuring adequate stability, scale-up and sterility.

Moreover, the safety of intraocular HGs has been evaluated in different animal models (mouse, rat, rabbit). Following FDA recommendations, particular attention has been paid to safety studies including hemocompatibility, carcinogenicity, cytotoxicity, genotoxicity, pyrogenicity, reproductive and theratogenic potential toxicities [188]. The evaluation of the intraocular pressure after injection, fundus examination by ophthalmoscope imaging, histological analysis and assessment of retinal function (by electroretinogram analyses) have confirmed that HGs are generally safe for intraocular use [137,189,190]. Nonetheless, so far, only a handful of HGs have been granted approval by the FDA to treat human diseases [138].

## 10. Conclusions

Therapeutic approaches to retinal diseases have the complexity of having to target the retinal tissues, when several ocular barriers have to be bypassed and drugs/therapeutic payloads have to diffuse across the vitreous mesh. Slow, sustained-release platforms such as in situ-forming HGs should make it easier to achieve the constant dose release of bioactive molecules over longer time periods, reducing the need of regular IVT injections and their potential unwanted side effects, including patient adherence to the treatment. More and more new designs are being tested to confront these issues, some with more success than others. Nonetheless, even though most of them are still at preclinical level, eventually, it is foreseen that some will successfully reach clinical trials, prompting the use of HG scaffolds as a new paradigm in intraocular therapies.

## Figures and Tables

**Figure 1 pharmaceutics-15-01484-f001:**
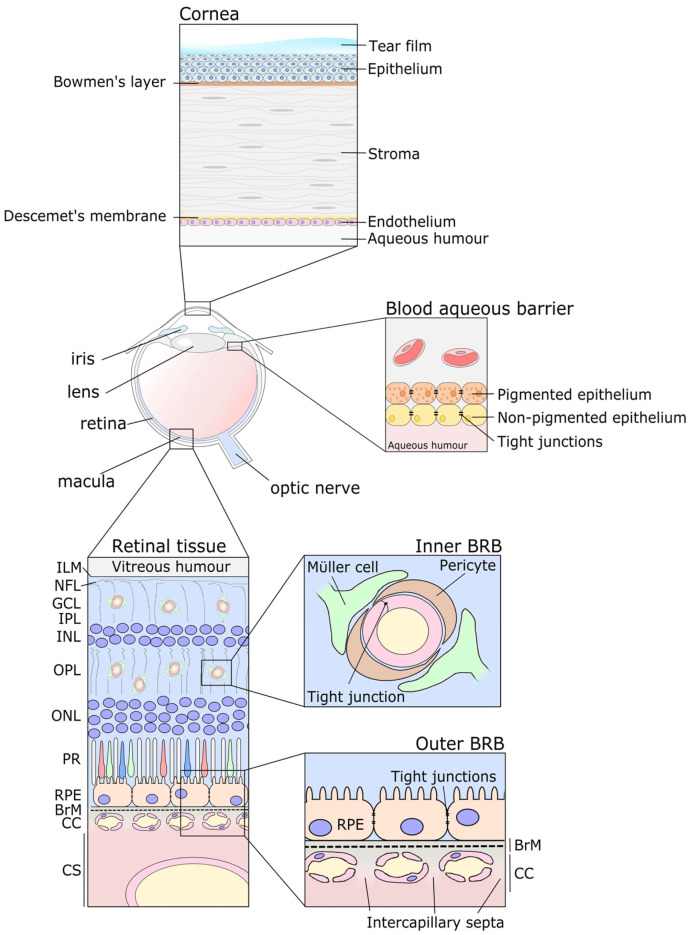
Schematic of different ocular barriers to drug delivery. The anterior chamber of the eye is protected by the transparent cornea, covered in the tear film, and which comprises the corneal epithelial layer, Bowman’s layer, the corneal stroma, Descemet’s membrane, and the corneal endothelium. The passage of material from plasma directly into the vitreous humor is regulated by the blood/aqueous barrier, where plasma nutrients and proteins from the ciliary body have to pass through both the pigmented and non-pigmented epithelial layers and their associated tight junctions. The retinal tissues in the posterior of the eye contain a number of different anatomical features, including the inner limiting membrane (ILM), nerve fiber layer (NFL), ganglion cell layer (GCL), inner plexiform layer (IPL), inner nuclear layer (INL), outer plexiform layer (OPL), outer nuclear layer (ONL), and photoreceptor (PR) cells. The photoreceptor cells are supported by their underlying retinal pigment epithelium (RPE), Bruch’s membrane (BrM), choriocapillaris (CC) and the choroidal stroma (CS). These structures are protected from the circulating blood flow by the blood/retinal barrier (BRB). The BRB is considered to comprise two separate parts: the inner BRB supplies nutrients to the inner retinal layers, and the outer BRB, which includes BrM and the RPE cell monolayer, symbolizes a significant barrier to accessing the outer retinal layers from the systemic blood circulation.

**Figure 2 pharmaceutics-15-01484-f002:**
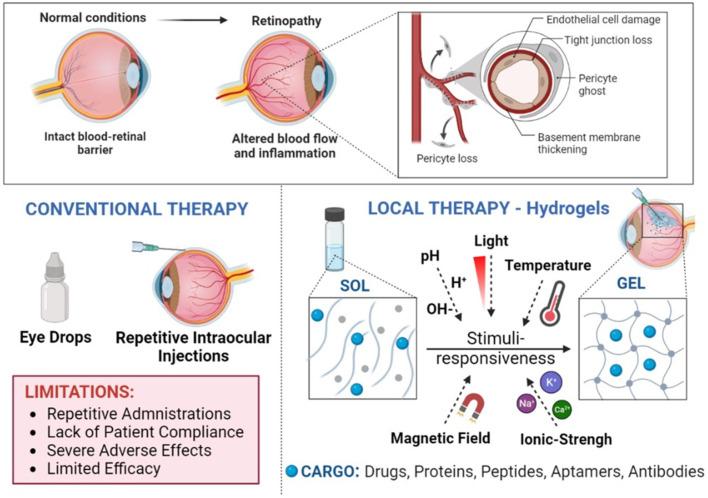
Graphical representation of inflammation and vascular alterations in retinopathies and the use of stimuli-responsive hydrogels as slow-release scaffolds for intraocular treatments.

**Figure 3 pharmaceutics-15-01484-f003:**
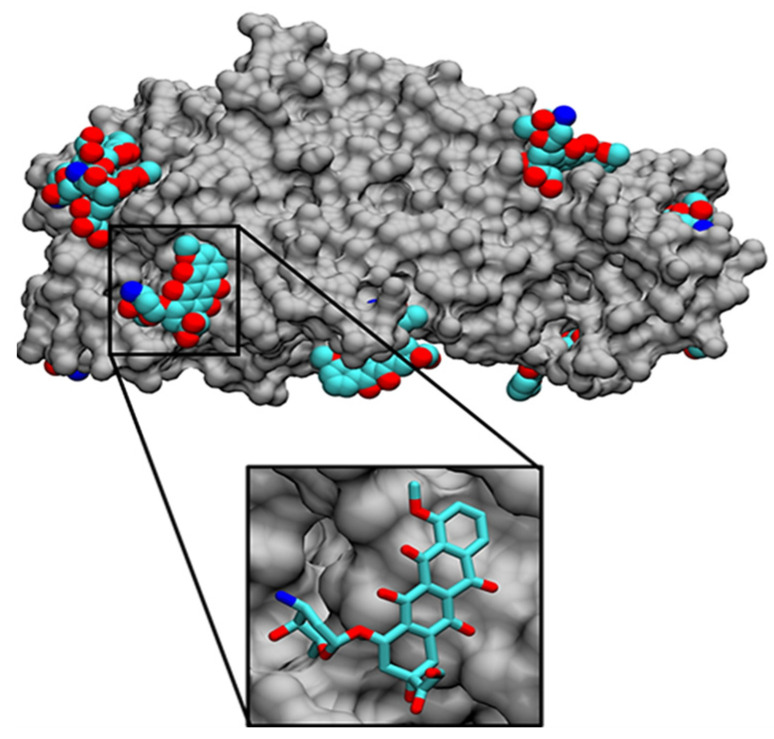
Doxorubicin interaction with the matrix of injectable HG based on a poly(N-isopropylacrylamide) derivative. Reprinted with permission from Elsevier [185]. Oxygen (Red); Nitrogen of the amine group (Blue); Carbon atoms (Turquoise).

**Table 1 pharmaceutics-15-01484-t001:** Table listing currently approved therapies employed for the treatment of selected retinal degenerative diseases.

Disease Indication	Drug/Brand Name	Drug Type	Mode of Action	Administration Route
AMD (wet) Diabetic retinopathy	Aflibercept Eylea	Fusion protein	Anti-VEGF	Monthly IVT injection
	Bevacizumab * Avastin	Monoclonal antibody		
	Brolucizumab Beovu	Single chain humanized antibody fragment (scFV)		
	Faricimab Faricimab-svoa	Bispecific IgG1 antibody		
	Pegaptanib Macugen	Pegylated aptamer		
	Ranibizumab Lucentis	Monoclonal antibody fragment (Fab)		
	Laser photocoagulation	-	Destruction of abnormal blood vessels	Laser surgery
Diabetic macula edema Posterior uveitis	Dexamethasone Ozurdex	Corticosteroid	Anti-inflammatory	Slow-release implant
	Fluocinolone acetonide Iluvien			
	Fluocinolone acetonide Retisert			
AMD (dry)	Pegcetacoplan SYFOVRE	Pegylated peptide	Complement inhibitor (C3)	Monthly IVT injection
	Avacincaptad pegol ** Zimura	Pegylated aptamer	Complement inhibitor (C5)	
IRD (biallelic *RPE65*)	Voretigene neparvovec Luxterna	Gene replacement therapy	RPE65 replacement	Subretinal injection AAV2

* Bevacizumab (Avastin) is FDA-approved for use in cancer treatments but is often used off-label to treat CNV in wet AMD. ** It is believed that Avacincaptad pegol (Zimura) will be presented to the FDA for approval by the end of 2023.

## Data Availability

Not applicable.

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
