# Peer review of "Delivery Systems in Ocular Retinopathies: The Promising Future of Intravitreal Hydrogels as Sustained-Release Scaffolds"

_pharmaceutics, 2023, doi:10.3390/pharmaceutics15051484_

Round 1

Reviewer 1 Report

This manuscript entitled “INJECTABLE HYDROGELS AS DELIVERY SYSTEMS FOR INTRAVITREAL USE IN OCULAR RETINOPATHIES” introduces the utilization of hydrogels, specifically temperature-responsive hydrogels, as carriers for delivering anti-angiogenic compounds via intravitreal injection. It examines their benefits and drawbacks for intraocular administration and highlights the latest developments in their use for treating retinal diseases. The structure of the article is relatively reasonable and the content is comprehensive. However, there are still some problems with this manuscript, and this manuscript can be accepted after a major revision in Pharmaceutics. However, the following comments should be addressed before publication.

1.   The title of the manuscript does not accurately reflect its content. While the title suggests a focus on intravitreal injectable hydrogels, a significant portion of the manuscript does not fall within this scope. For instance, Table 1 includes elyea, which is not relevant to injectable hydrogels. It is recommended to revise the title and ensure that the manuscript's content aligns with its title to avoid confusion for readers.

2.   The section "Advantages of using slow-release delivery systems in the vitreal cavity" is complex and could benefit from reorganization. To improve its organization, subheadings could be introduced. This would help to present the information in a more structured and comprehensible way.

3.   In part 5 “stimuli-responsive HGs in retinopathies”, microsphere, nanogel, and liposome do not belong to injectable hydrogels.

4.   Please give more introduction and examples of natural polymer-loaded drugs for the treatment of ocular retinopathies in this manuscript.

5.   Some abbreviations should be given their full name the first time they appear in a manuscript, eg HG, IVT.

6.   Please pay more attention to the writing. In part 2, “ant-VEGF” should be replaced by “anti-VEGF”. In part 6, “see Fig. X” should be replaced by “see Fig.2”.

7.   It is highly recommended to cite the following reference regarding an injectable hydrogel as a platform for neuroprotective combined therapies in treating retinal degenerative diseases.

López-Cano, J., A, S., Andrés-Guerrero, V., Tai, H., Bravo-Osuna, I., Molina-Martínez, I., Wang, W., Herrero-Vanrell, R.* ‘Thermo-responsive PLGA-PEG-PLGA hydrogels as novel injectable platforms for neuroprotective combined therapies in the treatment of retinal degenerative diseases’. PHARMACEUTICS, 2021, 13(2), 234, DOI: 1999-4923/13/2/234.

Minor editing of English language required.

Reviewer 2 Report

The review manuscript discusses the use of delivery systems to address ocular retinopathies.

The following have to be addressed before it can be considered.

1. The authors need to add a paragraph at the end of the introduction section to give a brief overview of the manuscript.

 2. The second section on ocular barrier is appropriate as it gives an overview of ocular drug delivery hurdles.

The third section is also appropriate.

 3. The authors have to distinguish between systems which are still experimental and those that have reached clinical trials.

As such the review can be re-organized as current treatment, systems under clinical trials, and systems still under in vitro trials.

They could dedicate a section on small molecules loaded injectable hydrogels.

 4. A new section on Challenges and Future perspectives should be added.

Information present in the last sections of the manuscript can be re-organized to fit into the new section.

This section should contain the critical analysis of the area based on the authors’ expertise.

 5. Conclusion section is to be added.

Minor grammar check required.

Reviewer 3 Report

Rafael et al. reviewed current advances in slow and long-lasting drug release technologies and pharmacokinetics to sustain prolonged efficacy using externally controlled hydrogels as delivery vehicles for intravitreal injection of anti-angiogenic molecules. The contents are informative and significant in the clinical treatment of AMD and DR. However, the manuscript needs additional effort to improve its clarity. The detailed comments are listed below.

  1. Table 1, Pegaptanib was explained as a pegylated oligonucleotide. This drug appears to be a pegylated aptamer as Avacincaptad pegol. If this is correct, it would be better to be consistent. Also, the authors only mentioned the injection of anti-angiogenic compounds, but Table 1 includes drugs targeting the complement system, a potent part of a host’s innate immune system. Please clarify this point.  
  2. In the section regarding AAV2-based therapy on page 8, the authors stated that the expression of transgenes was observed for 10 years in animals and 7.5 years in humans. In theory, AAV vectors do not integrate genes into the genome, but how have these somatic cells kept their expression for such a long time? Also, the authors mentioned that AAV could lead to an inflammatory response, but it can be controlled by lowering the doses of AAV. Please add appropriate reference studies regarding this point.  
  3. Page 13, “Nowadays, most…a short lifespan.” Please add references.
  4. Page 13, “An alternative…the retina.” Please add references. 
  5. Page 14, “Among them, …for drugs and proteins.” Please add references. 
  6. Page 15, “Of note, …far from predictable.” Please add references. 
  7. In “Stimuli-responsive HGs in retinopathies,” the authors reviewed the new approaches for controlling drug release in response to different external stimuli. The contents are informative and novel, but this reviewer found them too descriptive. Please consider including an additional figure to summarize this section so readers can easily follow. 

Round 2

Reviewer 1 Report

It is recommended to accept this manuscript after the revision.

Reviewer 2 Report

The authors have incorporated all suggested changes.

Good